# Comparison of Mechanical Properties of Three Tissue Conditioners: An Evaluation In Vitro Study

**DOI:** 10.3390/medicina59081359

**Published:** 2023-07-25

**Authors:** Marcin Mikulewicz, Katarzyna Chojnacka, Zbigniew Raszewski

**Affiliations:** 1Department of Dentofacial Orthopaedics and Orthodontics, Division of Facial Abnormalities, Medical University of Wroclaw, Krakowska 26, 50-425 Wroclaw, Poland; marcin.mikulewicz@umw.edu.pl; 2Department of Advanced Material Technologies, Faculty of Chemistry, Wroclaw University of Science and Technology, Smoluchowskiego 25, 50-372 Wroclaw, Poland; katarzyna.chojnacka@pwr.edu.pl; 3SpofaDental, Markova 238, 506-01 Jicin, Czech Republic

**Keywords:** tissue conditioners, phthalate-free alternatives, gelling time, adhesion, solubility, GC, Visco Gel, FITT, denture fabrication, biocompatibility

## Abstract

*Introduction:* Tissue conditioners have been widely used in various clinical applications in dentistry, such as treating inflamed alveolar ridges, temporarily relining partial and complete dentures, and the acquisition of functional impressions for denture fabrication. This study aimed to investigate the mechanical properties of the most prevalent tissue conditioner materials on the market, including Tissue Conditioners (TC), Visco Gel (VG), and FITT (F). *Materials and Methods*: The three tissue conditioners, TC, VG, and F, were assessed based on the parameters mentioned above. The following tests were performed based on the ISO 10139-1 and ISO 10139-2 requirements: Shore A hardness, denture plate adhesion, sorption, water solubility, and contraction after 1 and 3 days in water. Additional tests are described in the literature, such as ethanol content and gelling time. The tests were carried out by storing the materials in water at 37 °C for 7 days. *Results:* The gel times of all tested materials exceeded 5 min (TC = 300 [s], VG = 350 [s]). In vitro, phthalate-free materials exhibited higher dissolution in water after 14 days (VG = −260.78 ± 11.31 µg/mm^2^) compared to *F* (−76.12 ± 7.11 µg/mm^2^) and experienced faster hardening when stored in distilled water (*F* = 33.4 ± 0.30 Sh. A, VG = 59.2 ± 0.60 Sh. A). They also showed greater contractions. The connection of all materials to the prosthesis plate was consistent at 0.11 MPa. The highest counterbalance after 3 days was observed in TC = 3.53 ± 1.12%. *Conclusions*: Materials containing plasticizers that are not phthalates have worse mechanical properties than products containing these substances. Since phthalates are not allowed to be used indefinitely in medical devices, additional research is necessary, especially in vivo, to develop safe materials with superior functional properties to newer-generation alternatives. In vitro results often do not agree fully with those of in vivo outcomes.

## 1. Introduction

Tissue conditioners (TC) are dental materials used to address inflammation or pressure points on the alveolar ridge, temporary relining of partial and complete dentures, and the creation of functional impressions for denture fabrication or restoration. These soft relining materials can be divided into temporary and long-term solutions. Some requirements for both types of materials overlap [1]. An ideal material of this type should have the following characteristics: resistance to fungal and bacterial growth, low water absorption, increased color stability, stain resistance, tear resistance, strong bonding with the denture base, dimensional stability, low glass transition temperature, easy processing, long shelf life, biocompatibility, high energy dissipation, and good elasticity [2,3,4]. One of the main problems with TC is their susceptibility to colonization of microorganisms due to water solubility and degradation, which can exacerbate denture stomatitis [5].

Currently, TC are being enhanced with various types of additives that possess antibacterial properties. These additives include nature-based raw materials such as terpinene-4-ol and cinnamaldehyde [6], Cocos nucifera oil [7,8], lemongrass [9], as well as special fillers such as surface prereacted glass ionomer [10], zinc oxide nanoparticles [10], ZnOAg nanoparticles [11], and substances with drug properties such as cetylpyridinium chloride [12], poly(acryloyloxyethyltrimethyl ammonium chloride)-grafted chitosan [13], and bioactive glass [14].

However, it is important to note that these studies conducted in the laboratory phase have not resulted in commercially available products with such enhanced properties. Therefore, it is important to compare the existing materials so that the dentist can choose the best material for a particular clinical situation.

Typically, these materials consist of a mixture of powder and liquid that combine to form a soft gel [15,16]. The powder component is composed of finely ground poly (ethyl methacrylate polymer) with a granule size ranging between 30 and 40 microns. On the other hand, the liquid component is a mixture of ethanol and plasticizer. A physical reaction occurs when the plasticizer is absorbed into the polymer, and the presence of ethanol accelerates this process, significantly reducing the required time from several hours to mere minutes [2].

However, it is important to be aware that certain TC may contain harmful plasticizers, such as phthalates, which have been linked to adverse effects on the human endocrine system. Recent studies have indicated that prenatal exposure to phthalates is associated with adverse effects on neurodevelopment, including lower IQ, attention and hyperactivity problems, and poorer social communication. The effects of these substances on the adult body can damage the liver, kidneys, lungs, and reproductive system. To address these concerns, the Consumer Product Safety Improvement Act and its final rule in 2018 have banned eight phthalates in children’s products under federal legislation [17,18].

In response to safety concerns regarding phthalates, alternative plasticizers have been developed for use in TC. Citrate or sebacate-based plasticizers are among the new generation of materials that are currently available on the market. However, further research is still necessary to evaluate the safety and efficacy of these alternative materials [19,20].

The clinical shelf life of TC is relatively short, typically lasting from a few days to a week. Factors such as chewing forces, fluid consumption, and food intake contribute to the rapid leaching of plasticizers from the material. This process results in the hardening and degradation of TC over time. Additionally, the high sorption capacity of these materials leads to color changes and facilitates the quick colonization of various microorganisms, making it challenging to maintain proper hygiene when using soft relining materials. Routine cleaning with a toothbrush and toothpaste can easily remove or tear these materials [21,22].

The hypothesis of this study posits that phthalate-free alternatives for TC will exhibit physical properties that are comparable to or even superior to those containing phthalates.

## 2. Materials and Methods

The tests were conducted using the following materials: Visco Gel (Dentsply), Charlotte, NC, USA) which consists of 120 g of powder (transparent poly (ethyl methacrylate)), 90 mL of liquid (a mixture of ethanol and citrate-based plasticizer), and 15 mL of separator based on Vaseline oil. GC Tissue Conditioner includes 90 g of white PEMA powder, 90 g of liquid (composed of dibutyl sebacate and ethanol), and a 12 g coating agent (consisting of ethyl acetate and dissolved polymer). Kerr FITT from Kerr (Brea, CA, USA) contains liquid (ethanol, butyl phthalate, and butyl glycolate plasticizer), and 100 g of white powder based on poly (ethyl methacrylate). These materials were subjected to various mechanical tests, as shown in Figure 1, and tested materials are presented in Table 1.

Several mechanical tests were conducted for each material, following the standard ISO 10139-1:2018 Dentistry, Soft lining materials for removable dentures, with the number of samples specified by the standard [23].

The gelling time was determined by measuring the time it took for the mixture of powder and liquid to form a soft gel (which indicates the ease of use). Shore A hardness, which measures the material’s resistance to indentation, was evaluated for samples stored in water at 37 °C for 7 and 14 days. This test provides insights into the material’s ability to absorb masticatory forces and its long-term shelf life. Adhesion to the denture plate was tested by measuring the force required to separate the material from the denture plate, with higher ethanol content indicating more washing out and a shorter lifespan on the denture surface. Ethanol content was measured by drying the samples. Sorption and solubility in water were determined by measuring the change in mass of the samples when immersed in water. Contraction after 1 and 3 days of water immersion was measured by comparing the dimensions of the samples before and after immersion, providing information on the dimensional stability of the materials. The presence of plasticizers in the materials was performed using infrared spectrophotometry. An overview of all the test methods is presented in Figure 1.

### 2.1. Working a Gelling Time

The materials were mixed according to the manufacturer’s instructions. For example, in the case of Kerr FITT, 1.5 g of powder was mixed with 1 g of liquid in a plastic jar using a spatula. The container was then tilted to check if the material was flowing, and the working time of the material was considered to have ended at this point. The surface of the material was touched with a clean PE stick until it adhered to the stick’s surface, indicating the gelation time of the material. Time measurements were carried out using a calibrated stopwatch. All tests were conducted under laboratory conditions at a temperature of 23 °C. A total of 20 tests were performed for each measurement, resulting in a total of 60 samples. A description of this study can be found in Saeed et al. [24], Murata et al. [25].

### 2.2. Shore A Measurements

Samples for Shore A hardness were prepared using a 40 × 6 mm high metal cylinder form. The mold was covered with polyester foil and flat metal slabs on both sides. After a 30-min interval, the form was disassembled and the samples were removed. The initial measurement was conducted using a Shore A HD3000 durometer (Hildebrand, Oberboihingen, Germany) after 1 h. A total of 10 samples were prepared for each material, and each sample was measured four times from both sides (a total of 30 samples and 120 measurements).

Subsequently, the samples were placed in distilled water at 37 °C. After the designated period, the samples were dried, and new hardness tests were performed. Following the testing procedure, the samples were immersed in fresh distilled water and stored for an additional 7 days to simulate long-term use, as required by the ISO 10139-1:2018 Dentistry, Soft lining materials for removable dentures, Part 1: Materials for short-term use [23], and in accordance with previous studies by Saeed et al. [24] and Nowakowska-Toporowska et al. [1].

### 2.3. Ethanol Concentration

Samples weighing 2 g of liquid were stored at 37 °C until a constant mass was achieved. The determination of a constant mass was based on consecutive weight measurements using an analytical balance. If two successive measurements did not differ by more than 0.001 g, the concentration of ethanol was determined as the difference in mass before and after drying. A total of 10 samples were utilized for each material, resulting in a total of 30 samples. The Denver 300 balance (Denver Instruments, Paris, OH, USA), calibrated for accuracy, was employed for the testing process. For a detailed description of the test procedure, please refer to Murata et al. [25].

### 2.4. Sorption and Solubility

Following the mixing instructions, the materials were poured into a metal form with a diameter of 20 mm and a thickness of 2 mm. The form was then covered with polyester foil and metal slabs on both sides. After a 30-min interval, the form was opened, and the gel-shaped materials were extracted. For each solubility and sorption test, a total of 20 samples were prepared, resulting in a total of 60 samples.

One hour after removing the samples from the form, they were weighed (M1) and placed in covered plastic jars filled with distilled water. The first group of samples was stored for 7 days at 37 °C, while the second group was stored for 14 days under the same conditions. This was completed to simulate the clinical short-term or long-term use of the material.

After the storage period, the tissue conditioner was removed from the water, dried using paper, and weighed again (M2). Subsequently, the materials were transferred to a desiccator for drying until a constant mass was achieved (M3). For the testing process, a calibrated Denver 300 balance (Denver Instruments, Paris, OH, USA) was used.

Sorption was calculated using Equation (1), recommended by ISO 10139-1:2018 [23]:(1)$$Sorption=\frac{\left(M2-M1\right)}{M1}$$

Solubility was obtained from the differences, recommended by ISO 10139-1:2018 [21]:(2)$$Solubility=\frac{\left(M1-M3\right)}{M1}$$

For a detailed description of the test, see Saeed et al. [24].

### 2.5. Adhesion between Two Materials

To measure the adhesion force between the denture base material and the soft relining, a tensile test was performed. Samples of hot curing resins, specifically Superacryl Plus (Spofa Dental, Jicin, Czech Republic), were polymerized in metal forms with dimensions of 25 × 25 × 3.2 mm. After the curing process, the surface of the acrylic plate was coated with TC in a 10 × 10 mm area. A second acrylic plate was then placed on top to create a joint. The soft material was allowed to cure at room temperature for 24 h.

The following day, the samples were placed onto a tearing strength instrument, specifically the Shimadzu KN 50 (Shimadzu, Kyoto, Japan). The elongation speed of the breaking head was 5 mm/min. The samples were elongated at the connection point made with the soft material. The test concluded when the sample split into two pieces.

A total of 20 samples were prepared for each test, resulting in a total of 60 samples used. A detailed description of the test can be found in ISO 10139-2:2018 and Chladek et al. [26]. The scheme is shown in Figure 2.

### 2.6. Contraction

TC are clinically used for functional impressions; for this reason, they must be time stable. To measure the shrinkage of materials over specific time intervals, we used a method commonly used to measure changes in elastomeric impression materials. A specially calibrated metal block was used, featuring lines spaced 24.805 mm apart, each line measuring 50 μm. The samples were in the form of cylinders with a diameter of 30 mm and a thickness of 2 mm. The TC were mixed in accordance with the manufacturer’s recommendations and applied onto the metal block, which complied with ISO 10139-2:2016 Dentistry—Soft lining materials for removable dentures—Part 2: Materials for long-term use [23].

After 20 min, the materials were removed from the mold and placed in distilled water at 37 °C. Following 24 h, the distance between the lines on the samples was measured using a Karl Zeiss microscope (Zeiss, Jena, Germany) equipped with a measuring device capable of capturing the distance between the two reference lines (M2). The contraction of the material was determined using the following formula. The water was replaced with a fresh portion, and the process was repeated after 3 days. Each day, the water was replaced with new water.

A total of 20 samples were prepared for each material, resulting in a total of 60 samples. The contraction was calculated based on Equation (3), which is recommended by the ISO 10139-1:2018 standard [23] and described in Chladek et al. [26].
(3)$$Contraction=\frac{24.805-M2}{24.805}*100\%$$

### 2.7. Infrared Spectroscopy

The monomer samples were analyzed using the Nicolet ID7 apparatus (Thermo Scientific, Waltham, MA, USA). Each sample underwent 64 scans [27]. The test involved placing one drop of liquid onto the measurement window of the instrument and pressing it with a specialized glass to prevent the evaporation of volatile components. After the measurement, the slide was cleaned using enatol (Sigma Aldrich, Prague, Czech Republic). Each material sample was measured three times, resulting in a total of nine measurements. The identification of plasticizers was performed using a data library.

### 2.8. Statistical Analysis

Statistical analysis was performed with nine repetitions for each parameter, providing a robust dataset for the study. The data were expressed as mean ± standard error of the mean. A one-way analysis of variance (ANOVA) was performed to test for significant differences between the different tissue conditioner materials, including GC, Visco Gel, and FITT. A *p*-value < 0.05 was considered statistically significant. Posthoc tests were performed following the ANOVA analysis. GraphPad Software Inc., located in San Diego, CA, USA, was used for the statistical analysis. The use of ANOVA enables reliable determination of differences between the tested materials and ensures the statistical significance of the results.

## 3. Results

### 3.1. Working and Gelling Time

Table 2 shows the results of the working and gelling times for commercial products.

The results show that the GC Tissue Conditioner has the longest working time (162 ± 2 [s]), while FITT (mixing ratio 1.3:1) has the shortest working time and gel time (92 ± 3 [s]). The Visco Gel material has the longest gelling time (350 ± 8 [s]).

### 3.2. Hardness Shore A

Table 3 presents the changes in Shore A hardness after 7 and 14 days in distilled water at 37 °C.

Initially, the Shore A hardness of the tested materials was below 20 degrees (*F* = 11.0 ± 0.10 Sh. A, TC = 0.19.1 ± 0.30 Sh. A). After storage in distilled water, the hardness of all materials increased significantly (*F* = 38.1 ± 0.20 Sh. A, VG = 59.2 ± 0.60 Sh. A).

### 3.3. Ethanol Concentration

Table 4 shows the concentration of ethanol in the different products.

The highest concentration of ethanol was measured for the FITT [19.5%] and the lowest concentration was measured for the Visco-gel [11%]. However, it is important to note that the difference between these results was not statistically significant.

### 3.4. Adhesion between Materials

Table 5 presents the adhesion between the soft material and the denture base.

The study results indicate that no significant difference was observed in the connection between the various TC and the denture base. In all cases (100%), the separation of the sample occurred in an adhesive manner at the border between the soft material and the acrylic material. Figure 3 illustrates the surface of the acrylic plate after the tissue conditioner material was torn off.

### 3.5. Sorption and Solubility

Table 6 provides an overview of the sorption and solubility results for the tissue conditioners.

The sorption and solubility values obtained are expressed as negative values because, instead of absorbing water and increasing in mass, the materials dissolve rapidly. Consequently, the measured mass (M2) is smaller than the initial mass (M1).

Among the materials tested, FITT demonstrated the smallest changes in sorption and solubility, with a value of −76.12 ± 7.11 µg/mm^2^. In contrast, Visco Gel exhibited significant changes over time, with a value of −260.78 ± 11.31 µg/mm^2^.

### 3.6. Contraction

Table 7 presents the results of the contraction of the materials after 1 and 3 days of immersion in distilled water.

Immediately after gelling, a 50-micron line was visible in all tested materials. However, after 24 h, such lines were not detectable in Visco-gel. The surface of the material appeared to be overdried.

The smallest shrinkage was observed in the FITT material (mixing ratio 1.2:1) after 24 h (99.11 ± 1.2 [%]). The largest shrinkage occurred in the Tissue Conditioner after 3 days (96.47 ± 1.12 [%]). However, it is important to note that the difference between these results was not statistically significant.

### 3.7. Infrared Spectroscopy

The infrared scans and SDS analysis reveal the composition of the materials. The FITT material (Kerr) contains butyl phthalate and butyl glycolate, the Visco Gel material (Dentsply) contains acetyl tributyl citrate, and the Tissue Conditioner material (GC) contains dibutyl sebacate. The spectrum is shown in Figure 4.

For the FITT there are characteristic FT-IR peaks in the spectrum (cm^−1^): 1724.9 (carbonyl ester), 1280.9 and 1074.4 (–C–O–stretching), 742.46 (ortho disubstitution) (Blue color). Acetyl tributyl citrate exhibits characteristic peaks, such as the C=O absorption at 1738 cm^−1^, which is observed at slightly higher wavenumbers than the phthalate. Additionally, the strong asymmetric C–O stretching at 1182 cm^−1^ is typical of citrates (Pink color). The material from GC contains peaks at 2927–2852 cm^−1^, attributed to alkene (–CH_2_) groups, while intense peaks at 1159 and 1735 cm^−1^ are indicative of C–O and C=O formation, respectively.

## 4. Discussion

This study aimed to investigate the mechanical properties of three commercially available TC. However, the initial hypothesis was not confirmed as the materials containing nonphthalate plasticizers did not exhibit superior properties compared to the FITT material.

A very important feature of soft relining materials is their ability to absorb chewing forces in the oral mucosa following surgical treatment. Shore A hardness is a measurable parameter that indicates the material’s ability to cushion and relieve pressure in sensitive regions. An excessive increase in hardness compromises the material’s primary function. During storage in water, the materials experience an increase in hardness due to the leaching of ethanol and plasticizers. Similar findings of gradual hardness increase have been reported by other authors [2,22,28].

For soft and extra-soft materials, Shore A hardness values should be below 25 °C after 24 h of aging in distilled water at 37 °C. After 28 days of aging, Shore A hardness values should be below 55 °C [26]. The current study’s results align with previous studies demonstrating an increase in hardness over time for TC [29,30]. However, it is important to note that the magnitude of hardness increase varied among the tested materials. For instance, GC Tissue Conditioner exhibited an increase in hardness from 19.0 to 50 Shore A after 14 days, while FITT 1.2/1 increased from 11.1 to 33.4 Shore A during the same period. These findings emphasize the importance of material selection based on the desired hardness, which may vary depending on the specific clinical situation [31].

Ethanol is an important component in TC as it accelerates the gelling process of the material. However, it is known to be washed out in the oral environment within the first 24 h. In the conducted tests, the content of evaporated alcohol ranged from 11% for the Visco Gel material to 19.5% for the FITT material. Excessive alcohol content can lead to a positive reaction in a breathalyzer test shortly after denture relining [32].

The choice of gelling agent in TC is also crucial as it affects their overall performance. Researchers are exploring alternative gelling agents to improve long-term stability and performance [9,21,33,34]. Some attempts have been made to prepare materials without ethanol as a gelling agent.

GC Tissue Conditioner, for example, incorporates a special lacquer to reduce solubility and enhance the adhesion between the denture base and the relining material [35]. However, the results of the current study did not support these claims. The adhesion of soft lining materials to acrylic was found to be the same for all tested products (0.11 MPa), consistent with findings from other authors [36,37]. Wang et al. suggested that the adhesive strength between the denture plate and tissue conditioner may depend on the specific plasticizer used, as demonstrated in their study on various citrates [38].

Phthalates, which are known to have adverse effects on the human body, are a concern when in prolonged contact with materials like TC. They can lead to reproductive system changes, and their decomposition products can negatively affect the kidneys and lungs. Currently, FITT material contains phthalates, while Tissue Conditioner and Visco Gel utilize alternative plasticizers [18].

The results of the conducted tests clearly demonstrate that TC exhibit sorption (water uptake) and solubility after a relatively short period (7 days). However, after 2 weeks, these materials start to dissolve, and the amount of water absorbed becomes lower than the eluted plasticizer, resulting in negative sorption values. Among the tested materials, Visco Gel from Dentsply showed the highest degradation after 14 days (−260.78 ± 11.31 µg/mm^2^), compared to FITT (−76.12 ± 7.11 µg/mm^2^), which contains tributyl citrate according to the IR analysis. The citrate plasticizer with a straight chain structure is more easily eluted compared to the aromatic structure found in the Kerr material. A higher powder/liquid mixing ratio (lower plasticizer concentration) leads to less sorption and leaching for FITT. These findings are consistent with previous studies [2,9,24,26,38,39,40,41].

Another important application of TC is for making functional impressions. During this process, the material is applied under the denture base for a short period, typically ranging from 2 to 24 h. It gradually adapts to the oral cavity, undergoing deformation to ensure proper function. It is crucial for the material to maintain its dimensions during this period. The tested materials in this study exhibited low shrinkage, with less than 1% after 24 h and less than 2% after 3 days (FITT 1.5/1: 98.96 ± 1.14% after 1 day, 97.36 ± 1.26% after 3 days). However, the Visco Gel material showed significant dimensional changes after 24 h, and its surface appeared dry, making it difficult to observe the 50-micron lines. These results align with previous studies on materials such as Coe Super Soft and older formulas of Visco Gel containing phthalates [41].

The test results for tissue conditioner materials (GC, Visco Gel, and FITT) demonstrate differences in various properties, including working and gelling times, hardness, ethanol content, adhesion, sorption, solubility, and contraction. These differences can significantly impact the performance and properties of the materials, affecting factors such as gelling rate, solubility, and adhesion to the denture base [42]. It is crucial for dental professionals to understand these properties in order to choose the most appropriate tissue conditioner material for their patients. Furthermore, TC have the potential to serve as drug delivery systems and research efforts have been made to develop TC with antimicrobial properties [43,44,45,46,47,48].

### 4.1. Future Perspectives

This study provides valuable information on the mechanical properties of commercially available TC. However, further research is required to improve these materials and enhance their clinical performance. It is important to consider the biocompatibility of TC, particularly with the increasing use of phthalate-free alternatives. Researchers should incorporate findings on cytotoxicity and potential adverse effects of alternative plasticizers on oral mucosal cells into the evaluation and comparison of TC to ensure their safety and performance standards for clinical use [16]. Addressing the susceptibility of TC to microbial colonization is also important. The evaluation of alternative gelling agents is necessary since ethanol has drawbacks such as rapid evaporation and potential interference with breath analyzers. Optimizing material properties for patient-specific needs is essential, as different TC exhibit varying mechanical properties, which can be advantageous depending on the clinical situation. Pursuing these research directions has the potential to significantly enhance the clinical utility of TC, leading to better patient outcomes and overall satisfaction.

### 4.2. Clinical Significance

The clinical significance of phthalates in medical devices is an important issue that needs attention. The emergence of newer generation materials intended as phthalate alternatives could potentially display inferior functional properties when compared to traditional materials. Therefore, it is essential to conduct comprehensive research aimed at optimizing the properties and clinical performance of these novel materials [49,50]. For instance, a comparison can be made between acrylic and silicone materials, such as GC Reline II Soft, to determine their suitability.

## 5. Conclusions

➢The concentration of ethanol has a significant impact on the gelling time of TC, whereby higher concentrations result in shorter working and gelling times.➢TC with a higher alcohol content exhibit increased solubility.➢Straight-chain plasticizers, such as citrate, can be easily washed out of TC, leading to higher sorption and solubility of the materials.➢Lacquer presents an intriguing alternative for GC products, as it reduces the sorption of TC.➢Materials containing nonphthalate plasticizers demonstrate higher solubility and increased hardness in in vitro tests when stored in distilled water.➢Understanding the properties of commercial TC is essential for their optimal clinical performance.

## Figures and Tables

**Figure 1 medicina-59-01359-f001:**
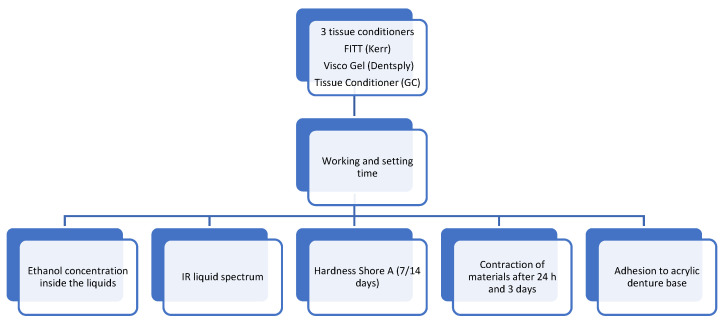
Graphical scheme of the tests carried out.

**Figure 2 medicina-59-01359-f002:**
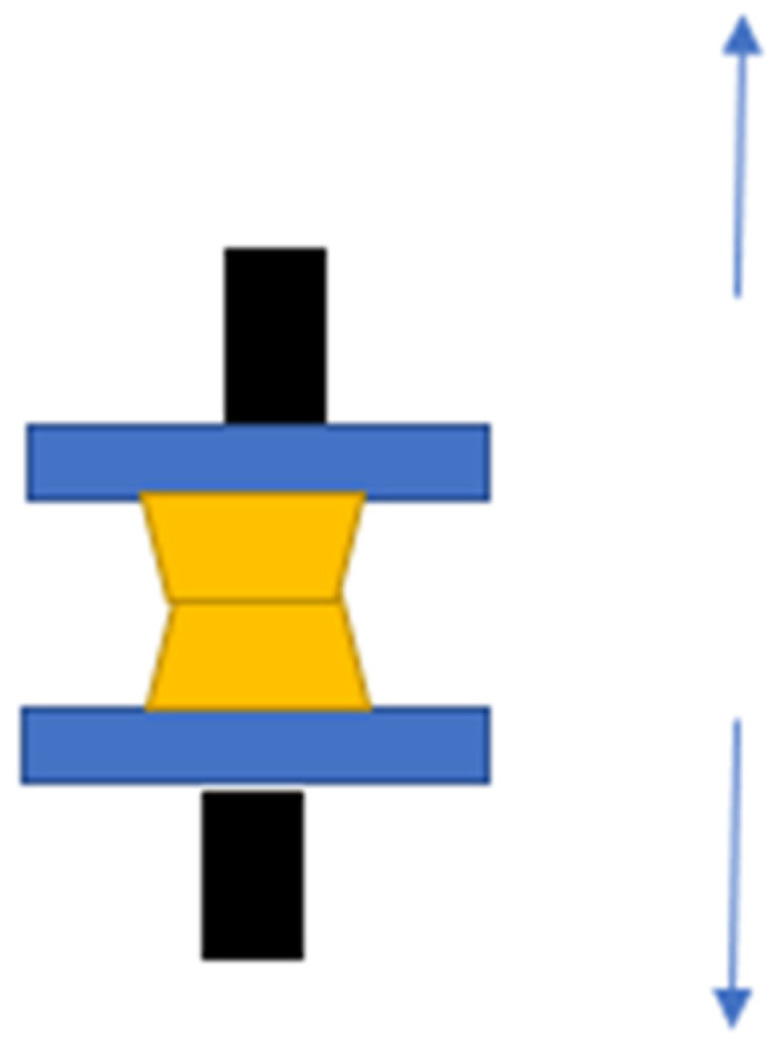
The samples and the breaking method (black attachment to the elongation machine, blue: acrylic plates, yellow: tissue conditioner).

**Figure 3 medicina-59-01359-f003:**
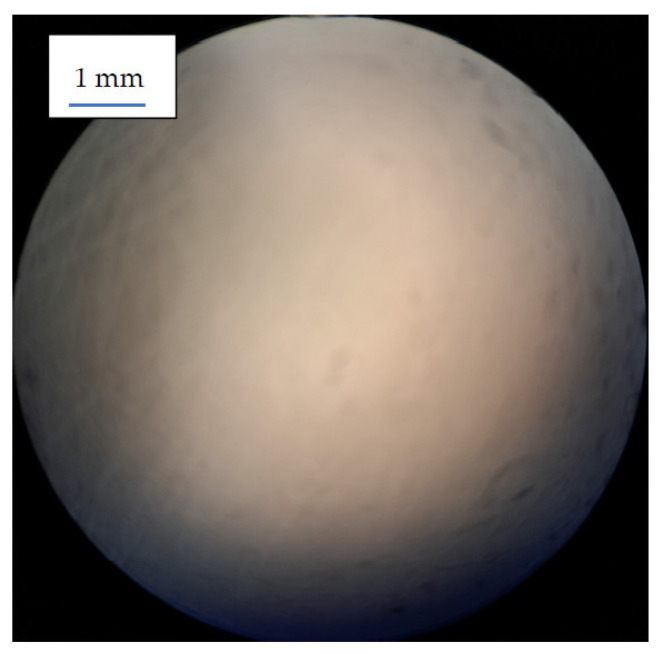
Acrylic disc after tearing off the tissue conditioners from their surface. The lack of traces of soft material proves the type of adhesive connection (magnification 25×).

**Figure 4 medicina-59-01359-f004:**
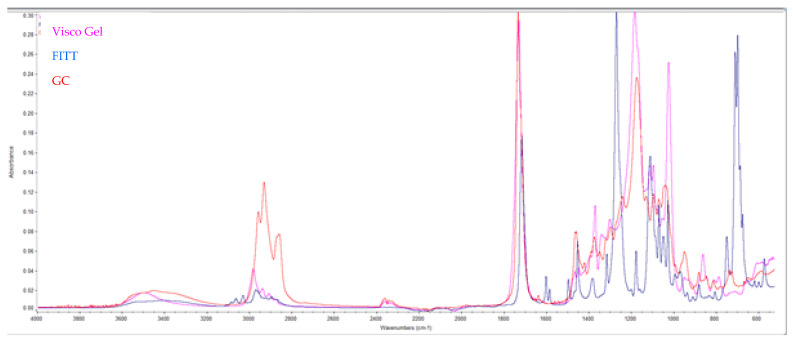
Infrared spectrum of the liquids of TC test materials.

**Table 1 medicina-59-01359-t001:** Materials and instruments used for testing.

Material	Producer	LOT
Tissue conditioner	GC (Tokyo, Japan)	1801121
FITT	Kerr (Scafati, Italy)	9529105
Visco Gel	Dentsply (Constance, Germany)	1805000667

**Table 2 medicina-59-01359-t002:** Working and gelling time of tissue conditioner materials.

Tissue Conditioner Material	Mixing Ratio (g)	Work Time (s)	Gel Time (s)
GC Tissue Conditioner	1.2:1	162 ± 2 *	302 ± 5 *
FITT	1.5:1	92 ± 3 *	231 ± 9 *
FITT	1.2:1	121 ± 4 *	300 ± 6 *
Visco Gel	1.5:1	149 ± 4 *	350 ± 8 *

* *p* value < 0.01.

**Table 3 medicina-59-01359-t003:** Changes in Shore A hardness after 7 and 14 days in distilled water at 37 °C of different tissue conditioner materials.

Composition	Initial Shore A Hardness (°)	Shore A Hardness after 7 Days (°)	Shore A Hardness after 14 Days (°)
GC Tissue Conditioner 1.2/1 *	19.1 ± 0.3 *p* = 0.01	28.6 ± 0.2 *p* = 0.04	51.3 ± 0.4 *p* = 0.01
FITT 1.5/1 *	12.7 ± 0.1	39.1 ± 0.3 *p* = 0.035	38.1 ± 0.2 *p* = 0.035
FITT 1.2/1 *	11.0 ± 0.1	13.3 ± 0.2 *p* = 0.048	33.4 ± 0.3 *p* = 0.047
Visco Gel 1.5/1 *	17.0 ± 0.1 *p* = 0.0087	40.0 ± 0.1 *p* = 0.0088	59.2 ± 0.6 *p* = 0.009

* Mixing ratio between powder and liquid [g].

**Table 4 medicina-59-01359-t004:** Ethanol concentration for different products.

Material	Ethanol (%)
GC Tissue Conditioner	12
Visco Gel	11
FITT	19.5

**Table 5 medicina-59-01359-t005:** Adhesion between soft material and denture base, by the tensile test of two acrylic pieces joined by a tissue conditioner.

Material	Adhesion (MPa)
GC Tissue Conditioner 1.2/1 *	0.110 ± 0.013
FITT 1.2/1 *	0.117 ± 0.009
FITT 1.5/1 *	0.105 ± 0.006
Visco-gel 1.5/1 *	0.110 ± 0.007

* Mixing ratio between powder and liquid [g].

**Table 6 medicina-59-01359-t006:** Results from the sorption and solubility of commercial products.

Material	Sorption after 7 Days (µg/mm^2^)	Solubility after 7 Days (µg/mm^2^)	Sorption after 14 Days (µg/mm^2^)	Solubility after 14 Days (µg/mm^2^)
GC Tissue Conditioner 1.2/1 *	30.18 ± 3.45	−23.31 ± 4.15	−102.11 ± 2.5	−142.19 ± 3.00
GC Tissue Conditioner 1.2/1 * with lacquer	26.66 ± 5.55	−23.06 ± 2.88	−102.51 ± 2.27	−132.73 ± 3.12
FITT 1.5/1 *	39.36 ± 4.39	−10.27± 2.71	−23.37 ± 2.36 *p* = 0.029	−76.12 ± 7.11 *p* = 0.0096
FITT 1.5/1 *	43.55 ± 5.11	−13.78± 3.82	−44.68 ± 3.06 *p* = 0.048	−106.52 ± 3.27 *p* = 0.099
Visco Gel 1.5/1 *	33.15 ± 2.32	−27.45 ± 4.32	−1980.59 ± 9.88 *p* = 0.0001	−260.78 ± 11.31 *p* = 0.001

* Mixing ratio between powder and liquid [g].

**Table 7 medicina-59-01359-t007:** Contraction of materials [%] stored in distilled water for a period of 1 and 3 days.

Material	24-h Contraction (%)	3 Days of Contraction (%)
FITT 1.2/1 *	99.11 ± 1.2	97.39 ± 1.05
FITT 1.5/1 *	98.96 ± 1.14	97.36 ± 1.26
GC Tissue Conditioner 1.2/1 *	97.63 ± 1.33	96.47 ± 1.12
Visco-gel 1.5/1	There is no possibility of seeing the lines.	There is no possibility of seeing the lines.

* Mixing ratio between powder and liquid [g].

## Data Availability

Data sharing is not applicable to this article.

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
