# Peer review of "Comparison of Mechanical Properties of Three Tissue Conditioners: An Evaluation In Vitro Study"

_medicina, 2023, doi:10.3390/medicina59081359_

Round 1

Reviewer 1 Report

1. Extensive English language edits is requested

2. Abstract not informative and not well presented.

3. Redundancy evident in literature review . Paragraphs not sequential. 

4. Methodology not well presented,  no adequate citations provided for many procedures used, sample size, specimens dimensions. 

5. Results not focused, looks as chopped from a thesis and not for a scientific journal.  No data appear in text. Explanation provided which should be in discussion  section

6. Again Redundancy evident with repetitions of many Results Paragraphs 

7. Conclusions and clinical significance are similar and not focusex

1. Extensive English language edits .Difficult fo follow 

Author Response

Dear Professor, thank you very much for your positive review of our article. All your suggestions have been added to the text.

Suggestion 1

Extensive English language edits is requested

The English language has been improved

Suggestion 2

Abstract not informative and not well presented.

Abstract has been corrected, numerical values of obtained results have been introduced, Conclusion contains only the most important information

Suggestion 3

Redundancy evident in literature review . Paragraphs not sequential

Part of the introduction has been shortened; the information has been arranged so that it appears in a logical order

Suggestion 4

Methodology not well presented,  no adequate citations provided for many procedures used, sample size, specimens dimensions

The Materials and Methods section has been modified, with the addition of flickering references, sample size, and the number of samples used for each test.

Suggestion 5

Results not focused, looks as chopped from a thesis and not for a scientific journal.  No data appear in text. Explanation provided which should be in discussion  section

The Results part has been modified, added numeric values to the text

Suggestion 6

Again Redundancy evident with repetitions of many Results Paragraphs.

The Results section has been rearranged and the paragraphs are presented in a logical order. It has also been shortened to the most important information.

Suggestion 7

Conclusions and clinical significance are similar and not focusex

Parts of theses have been modified and separated so that the conclusions contain research information and clinical significance information for end users.

We are grateful to the Reviewer and the Editor for all the remarks and suggestions that helped to improve our manuscript. We have included additional paragraphs to the revised manuscript according to the recommendations. We hope that with these revisions, our manuscript serves better the readers of the Journal than the initially submitted version would, and once more, we thank the Reviewers for their work

Authors

Reviewer 2 Report

Dear authors,

I want to congratulate with you for such nice work you’ve done with this paper. I think that it is a good idea to test all these properties of soft conditioning materials. Your method seems to be correct and precise in their main aspects. But there are some suggestions to improve the quality of your work. 

Line 62: what do you mean putting “good elasticity” next to “low modulus of elasticity”: it seems to be a contradiction. Please specify better this period.

Please clarify better which are the adverse effects of phthalate on human body in the introduction

Please report reference lines 172-174 thanks

In the results is missing the statistical analysis: please report each p value and CI interval of your analysis in the respective tables

In the discussion section please argue completely the toxicity of phthalate in human organism, which dental material contains it and the alternatives that dental industries have developed to substitute it.

Please provide more updated references: there are too many old references that may result inappropriate. In these years of scientific progression perhaps have been carried out more interesting studies with latest materials.

Moreover, I suggest to you to cite and consider the existence of “GC RELINE” which is a polyvinylsiloxane for definitive conditioning, even in future studies you may carry out.

The last comment which can improve your valuable work, is to change the title because may sounds misleading: this is a comparation between three soft conditioning materials. One of three contains butyl phthalate (FITT) so you can’t talk about phthalate-free tissue conditioners.

English could be improved: there are many sentences difficult to understand. Readability have to be improved.

Author Response

Dear Professor, thank you very much for your positive review of our article. All your suggestions have been added to the text.

suggestion 1

Line 62: what do you mean putting “good elasticity” next to “low modulus of elasticity”: it seems to be a contradiction. Please specify better this period.

The sentence has been corrected. These materials should be characterized by high flexibility

long shelf life, biocompatibility, high energy dissipation, good elasticity.

Suggestion 2

Please clarify better which are the adverse effects of phthalate on human body in the introduction.

This part has be modified:

Recent studies  show that prenatal exposure to phthalates is associated with adverse impacts on neurodevelopment, including lower IQ, and problems with attention and hyperactivity, and poorer social communication. The effects of these substances on the adult body can damage the liver, kidneys, lungs, and reproductive system. Consumer Product Safety Improvement Act and its final rule in 2018, defines that eight phthalates are  banned in children's products through this federal legislation.

Suggestion3

Please report reference lines 172-174 thanks

It has been added ref number [19]

Suggestion 4.

In the results is missing the statistical analysis: please report each p value and CI interval of your analysis in the respective tables

Statistical analysis result values have been added

Suggestion 5

In the discussion section please argue completely the toxicity of phthalate in human organism, which dental material contains it and the alternatives that dental industries have developed to substitute it.

This part has been modified to add the effect of phthalates on the human body

As mentioned at the beginning, phthalates can have a very adverse effect on the human body. During prolonged contact with materials such as Tissue Conditioners, they can cause changes in the reproductive system and their decomposition products adversely affect the kidneys and lungs. Currently, FITT material contains phthalates and Tissue Conditioner and Visco Gel, alternative plasticizers.

Suggestion 6

Please provide more updated references: there are too many old references that may result inappropriate. In these years of scientific progression perhaps have been carried out more interesting studies with latest materials.

We have added the latest testimonials from the last years to our article. However, by quoting older works, we wanted to show how complex the problem of tissue conditioners is, and how long scientists and researchers have been trying to solve it.

Suggestion 7

Moreover, I suggest to you to cite and consider the existence of “GC RELINE” which is a polyvinylsiloxane for definitive conditioning, even in future studies you may carry out.

Is a good idea for further research, We will try to take this into account in our next research.

 Suggestion 8

The last comment which can improve your valuable work, is to change the title because may sounds misleading: this is a comparation between three soft conditioning materials. One of three contains butyl phthalate (FITT) so you can’t talk about phthalate-free tissue conditioners.

Title has been changed:

Comparison of mechanical properties of three tissue conditioners: an evaluation in vitro study

We are grateful to the Reviewer and the Editor for all the remarks and suggestions that helped to improve our manuscript. We have included additional paragraphs to the revised manuscript according to the recommendations. We hope that with these revisions, our manuscript serves better the readers of the Journal than the initially submitted version would, and once more, we thank the Reviewers for their work

Authors

Round 2

Reviewer 1 Report

The quality of the revised version is improved, however, authors should try to combine English edits and writing with scientific terminology. Paragraphs are still in inappropriate locations. In the abstract, the methodology is brief as authors refer the reader to the introduction.  Each section has to stand alone clear and with sufficient details. 

The methodology in the main text is improved but there is still no adequate citations for all procedures carried out. ISO standards were added but this is not sufficient.  References from previous studies need to be included as citations.

The results need to be rewritten in a scientific paper style without providing description or explanation 

Conclusions are still general and not specific or based on the results.  Similar to the test hypothesis. 

Much work has to be done on this manuscript 

Improved but edits needed and combined with the use of appropriate scientific terminology  especially in methodology and results sections

Author Response

Dear Professor, thank you very much for your positive review of our article. All your suggestions have been added to the text.

Suggestion 1

the quality of the revised version is improved, however, authors should try to combine English edits and writing with scientific terminology. Paragraphs are still in inappropriate locations. In the abstract, the methodology is brief as authors refer the reader to the introduction.  Each section has to stand alone clear and with sufficient details.

The manuscript has been revised to our best intentions. The Abstract part, and the main text. Current fixes are marked in green. Appropriate literature references have been added to the materials and methods section. Most of the tests are precisely described in the ISO standard, and as such were used in the tests. What we describe.

Suggestion 2

The methodology in the main text is improved but there is still no adequate citations for all procedures carried out. ISO standards were added but this is not sufficient.  References from previous studies need to be included as citations.

Appropriate references have been added in the method description.

Suggestion 3

The results need to be rewritten in a scientific paper style without providing description or explanation

The results were limited to presenting them without comment, this part is included in the discussion

Suggestion 4

Conclusions are still general and not specific or based on the results.  Similar to the test hypothesis.

Conclusion has been corrected.

  In order to maintain the highest standards of the journal, we decided to submit our manuscript to the office of experts specializing in scientific texts in English. So the work has been reviewed and corrected once again by good experts. We will send the appropriate certificate in the attachment.

Reviewer 2 Report

Dear authors,

Thank you for following and addressing my suggestions. Now paper has more scientific soundness. 

The last thing I suggest to you is to report better in respective tables the values of p. For example, you assumed that you considered significative a value of p < 0.05 (as usual). But in the tables you have to report the exact value of p (p = 0.0028 for example). If it is minor of 0.05 you can place an asterisk to indicate that it is significant. This is the only thing to correct, to be rigorous. 

Thank you.

English is fine

Author Response

Dear Professor, thank you very much for your positive review of our article. All your suggestions have been added to the text.

Comment 1

Thank you for following and addressing my suggestions. Now paper has more scientific soundness.

  • Thank you very much

Comment 2

The last thing I suggest to you is to report better in respective tables the values of p. For example, you assumed that you considered significative a value of p < 0.05 (as usual). But in the tables you have to report the exact value of p (p = 0.0028 for example). If it is minor of 0.05 you can place an asterisk to indicate that it is significant. This is the only thing to correct, to be rigorous.

  • The individual p< values have been added to table. thank you